# StatQAT: Statistical Quantizer Optimization for Deep Networks

## Abstract

Quantization is essential for reducing the computational cost and memory usage of deep neural networks, enabling efficient inference on low-precision hardware. Despite the growing adoption of uniform and floating-point quantization schemes, selecting optimal quantization parameters remains a key challenge, particularly for diverse data distributions encountered during training and inference. This work presents a novel statistical error analysis framework for uniform and floating-point quantization, providing theoretical insight into error behavior across quantization configurations. Building on this analysis, we propose iterative quantizers designed for arbitrary data distributions and analytic quantizers tailored for Gaussian-like weight distributions. These methods enable efficient, low-error quantization suitable for both activations and weights. We incorporate our quantizers into quantization-aware training and evaluate them across integer and floating-point formats. Experiments demonstrate improved accuracy and stability, highlighting the effectiveness of our approach for training low-precision neural networks. [1]

## 1 Introduction

Quantization is a key optimization for deploying large deep learning models efficiently (1). By reducing parameter precision, quantization decreases model size and improves memory bandwidth utilization through reduced data movement (2; 3). When activations are also quantized, forward-pass computations can leverage specialized low-precision hardware units such as INT8, FP8, and FP4 matrix multipliers, which are widely used in linear and convolution layers (4; 5). These optimizations collectively enable low-latency, high-throughput inference (4).

Quantization schemes are typically categorized as non-uniform, uniform, or floating-point. Non-uniform quantization offers maximum flexibility by allowing arbitrary placement of quantization levels, but its hardware incompatibility limits its use primarily to weight-only quantization (6). Uniform quantization constrains levels to be equally spaced, enabling efficient implementation with integer formats such as INT8/4/2 or UINT8/4/2 (7; 8). Floating-point quantization, such as FP8 or FP4, follows the standard floating-point representation, where levels are dense near zero and expand exponentially, making it well-suited for parameters clustered around zero (9; 10).

In this work, we introduce a statistical framework for analyzing and minimizing quantization error in uniform and floating-point schemes. Unlike classical quantizer design methods that optimize parameters offline for fixed distributions, our framework adapts quantizer parameters during training to account for dynamically evolving weight and activation distributions. Our approach enables principled selection of quantization parameters for both weights and activations based on the underlying data distribution. We propose iterative quantizers tailored to arbitrary data distributions commonly observed in activations, and analytic quantizers optimized for approximately Gaussian-distributed weights, a common assumption during training (11). We focus on quantization-aware training (QAT) to evaluate these quantizers across integer and floating-point formats. Importantly, our analysis extends to floating-point formats such as FP4, which are gaining native hardware support in modern accelerators such as NVIDIA Blackwell. To the best of our knowledge, this work provides one of the first error-driven analyses of floating-point quantization in the context of QAT, deriving closed-form analytic updates that improve both theoretical understanding and practical hardware relevance.

Our contributions are summarized as follows:

---

[1]Code is submitted with the paper. Upon acceptance, it will be publicly available.

- We introduce a statistical framework for analyzing quantization error in both uniform and floating-point quantization schemes, enabling principled selection of quantizer parameters based on the underlying data distribution.

- We develop iterative quantizers for arbitrary activation distributions and analytic quantizers for approximately Gaussian weight distributions, allowing efficient optimization of quantization parameters during training.

- We extend the statistical error analysis to floating-point formats and derive closed-form updates that enable practical integration with quantization-aware training.

- Through experiments across integer and floating-point formats, we demonstrate improved quantization behavior and practical relevance for modern low-precision hardware.

## 2 Background

The quantization operator is a many-to-one function, formulated in its general form as follows:

$$Q(x|\boldsymbol{l}, \boldsymbol{t}) = l_k, \quad \text{if } t_k < x \le t_{k+1}, \tag{1}$$

where $x \in \mathbb{R}$ denotes a scalar input to be quantized, $\boldsymbol{t} \in \mathbb{R}^N$ is a list containing $N$ sorted thresholds defining the predetermined quantization intervals, and $\boldsymbol{l} \in \mathbb{R}^{N-1}$ contains corresponding quantization levels for each interval. Given $x$, the operator compares its value to $N-1$ intervals defined by $N$ thresholds $\boldsymbol{t}$ and assigns it to the corresponding level $l_k$.

### 2.1 Non-uniform quantization

The non-uniform quantization scheme does not constrain the choice of levels and thresholds, making it highly flexible. There are $2N - 1$ parameters, including thresholds and levels. In practice, thresholds are often chosen as the midpoints between consecutive levels, such as $t_k = (l_k + l_{k-1})/2$ for $k = 1 : N - 2$, and $t_0 = -\infty$, $t_{N-1} = \infty$, to cover the full input range. This choice minimizes the mean-squared quantization error given a fixed set of levels.

### 2.2 Uniform quantization

In this scheme, $\boldsymbol{l}$ is constrained to have evenly spaced $N - 1$ quantization levels, which can be determined using a scale parameter $s$ and a shift parameter $z$. The levels and thresholds are then defined as: $l_k = sk + z$ for $k = 0 : N - 2$ and $t_k = s(k - 1/2) + z$ for $k = 1 : N - 2$. This scheme underlies integer quantization methods such as INT8, INT4, and INT2, which are widely deployed in hardware-efficient inference engines (4; 3).

When $z$ is set to $-(N/2 - 1)s$, the levels are symmetric around zero: $l_k = [-(N/2 - 1)s, \ldots, (N/2 - 1)s]$, and the thresholds become $\boldsymbol{t} = [-\infty, -(N - 3)s/2, \ldots, (N - 3)s/2, \infty]$. This symmetric setting reduces parameter count at the expense of increased quantization error. Appendix A provides implementation details of uniform quantization.

### 2.3 Float quantization

A float number comprises a sign bit, $E$ exponent bits, and $M$ mantissa bits. Its value is computed as $(-1)^s \times 2^{(e-b)} \times (f_0 + \sum_{m=1}^{M} f_m 2^{-m})$, where $b$ is a bias term for the exponent. $E$ bits define the exponent value $e$, and $\{f_m\}_{m=1:M}$ define the fraction. This representation introduces non-uniform spacing, where smaller values are represented with higher resolution, making float formats such as FP8 and FP4 particularly suited for data clustered around zero (10; 11).

The floating-point standard includes subnormal and normal regions (12). When $f_0 = 0$, subnormal representation is active to handle very small magnitudes. When $f_0 = 1$, the normal grid is active. The positive half of the grid points are denoted as:

$$g_p = \begin{cases} p2^{1-M-b}, & \text{if } p \in [0, 2^M - 1] \\ 2^{\lfloor \frac{p}{2^M} \rfloor - b}(1 + (p \bmod 2^M)2^{-M}), & \text{if } p \in [2^M, 2^{M+E} - c - 1] \end{cases} \tag{2}$$

for $p = 0 : 2^{M+E} - c - 1$ where $c$ accounts for special values like NaNs or INFs. This grid doubles its spacing every $2^M$ points. Including the negative half, the total number of grid points becomes $2^{M+E+1} - 2c - 1$. With scale $s$ and shift $z$ parameters, the quantization levels $l$ are defined as

$$l_k = \begin{cases} z - sg_{-k+2^{M+E}-c-1}, & \text{if } k \in [0, 2^{M+E} - c - 2] \\ z + sg_{k-2^{M+E}+c+1,}, & \text{if } k \in [2^{M+E} - c - 1, 2^{M+E+1} - 2c - 2] \end{cases} \tag{3}$$

where $s$ and $z$ are parameters to be determined. See Appendix B for the practical implementation.

## 3 Error-Driven Quantizer Parameter Optimization

The previous section introduced uniform and floating-point quantizers, each parameterized by a scale and shift to define quantization levels and thresholds. We now formulate the quantization error analytically for both schemes and propose iterative and analytic quantizers to optimize these parameters. For completeness, we include optimization details for non-uniform quantizers in Appendix G.

### 3.1 Uniform Quantizers: Error Model and Optimization

In the uniform quantization scheme, levels are defined as $l_k = sk + z$ for $k = 0 : N - 2$, and thresholds are $t_k = s(k - \frac{1}{2}) + z$ for $k = 1 : N - 2$, with $t_0 = -\infty$ and $t_{N-1} = \infty$. The mean-squared quantization error function is given by:

$$\mathbb{E}[e^2] = \mathbb{E}[(x - Q(x|\boldsymbol{l}, \boldsymbol{t}))^2] = \sum_{k=0}^{N-2} \int_{t_k}^{t_{k+1}} (x - l_k)^2 p(x) dx, \tag{4}$$

where the errors corresponding to each quantization level are integrated over $N - 1$ intervals given the data distribution $p(x)$. We decompose this function as the sum of two terms, clipping $E_c$ and stepping $E_s$ error functions. Clipping error function is defined as the error due to the saturated regions:

$$E_c = \int_{-\infty}^{-s/2+z} (x - z)^2 p(x) dx + \int_{s(N-5/2)+z}^{\infty} (x - s(N-2) - z)^2 p(x) dx, \tag{5}$$

whereas the stepping error function $E_s$ is defined as the power of stepping error $e_s = x - Q(x|\boldsymbol{l}, \boldsymbol{t})$, which is a random variable uniformly distributed on the interval $[-\frac{s}{2}, \frac{s}{2}]$ according to the stochastic uniform error model (13; 14). Thus $p(e_s) = \frac{1}{s}\mathbb{I}(-s/2 \leq e_s \leq s/2)$ with the mean value being zero due to symmetry. The mean squared value of $e_s$, which corresponds to the stepping error function $E_s$, is given by $\frac{s^2}{12}$. The total quantization error is $\mathbb{E}[e^2] = E_c + E_s$. Since reducing $s$ shrinks $E_s$ but increases $E_c$, this tradeoff defines an optimization problem over $s$ and $z$.

#### 3.1.1 Iterative Uniform Quantizer

When there is no prior information about the distribution of the data, we minimize Eq. 4 via alternating optimization over $s$ and $z$. Taking the derivatives of the error and setting them to zero yields:

$$s = \frac{\sum_{k=0}^{N-2} k \int_{t_k}^{t_{k+1}} (x - z) p(x) dx}{\sum_{k=0}^{N-1} k^2 \int_{t_k}^{t_{k+1}} p(x) dx}, \tag{6}$$

$$z = \frac{\sum_{k=0}^{N-2} \int_{t_k}^{t_{k+1}} (x - sk) p(x) dx}{\sum_{k=0}^{N-1} \int_{t_k}^{t_{k+1}} p(x) dx}. \tag{7}$$

Then, the thresholds are updated as $t_k = s(k - 1/2) + z$ for $k = 1 : N - 2$ where $t_0 = -\infty$ and $t_{N-1} = \infty$. First, the data points are quantized given the previous values of $s$ and $z$, and then, $s$ and $z$ are updated given the quantized data. This process resembles a modified 1D $k$-means algorithm, constrained so that cluster centers (quantization levels) are evenly spaced.

### 3.1.2 Normal-Optimal Uniform Analytic Quantizer

Assuming the data is normally distributed (a reasonable prior for weights due to $L_2$ norm regularization (11; 15)), we derive an analytic uniform quantizer for parameter quantization. The input space is unbounded when the data distribution is normal. Hence, a clipping point is determined, which results in a clipping error. Denoting the clipping point by $C$, the clipping error function for zero-mean normally distributed data is computed analytically as follows:

$$E_c = 2(\sigma^2 + C^2)\mathcal{Q}\left(\frac{C}{\sigma}\right) - 2C\frac{\sigma}{\sqrt{2\pi}}\exp\left(-\frac{C^2}{2\sigma^2}\right), \tag{8}$$

where $\mathcal{Q}(z) = \frac{1}{\sqrt{2\pi}}\int_z^\infty \exp^{-\frac{u^2}{2}}du$ is the $\mathcal{Q}$ function (see Appendix E for the derivation). Given a $C$ value, the step size is computed as $s = \frac{2C}{N-1}$ due to uniformity. Given that the stepping error is uniformly distributed, the stepping error function is given by

$$E_s = \frac{C^2}{3(N-1)^2}. \tag{9}$$

The goal is to maximize the signal-to-noise ratio:

$$SNR = \frac{\sigma^2}{E_c + E_s} = \left[2(1 + \frac{C^2}{\sigma^2})\mathcal{Q}\left(\frac{C}{\sigma}\right) - \frac{C}{\sigma}\sqrt{\frac{2}{\pi}}e^{-\frac{C^2}{2\sigma^2}} + \frac{C^2}{3\sigma^2(N-1)^2}\right]^{-1} \tag{10}$$

Given $N$ and $\sigma$, finding $C$ that maximizes SNR does not have a closed form, so we perform a numerical search. Different from the iterative method, this search is done offline without requiring any data. In practice, we set $\sigma^2 = 1$ and find the optimal clipping $C_{opt}$ that maximizes SNR with numerical search given the total number of levels $N$. Then, during the quantization process, the mean and variance of the the data are computed, and the scale and shift parameters are determined accordingly as $s = 2C_{opt}\sigma_{(x)}/(N-1)$ and $z = -s(N/2 - 1) + m_{(x)}$ where $\sigma_{(x)}$ and $m_{(x)}$ are empirical standard deviation and mean of the data points, respectively.

### 3.2 Floating point quantizers: Error Model and Optimization

Floating-point quantization introduces exponentially spaced levels that better match the distributions centered around zero (9; 10; 11). The quantization error function for floating-point schemes is analogous to the uniform case (Eq. 4), but uses the level definitions from Eq. 2, and sets the number of levels as $N = 2^{M+E+1} - 2c - 1$.

As in uniform quantization, the total mean-squared quantization error is decomposed into a clipping error $E_c$ and a stepping error $E_s$. The clipping error $E_c$ is defined identically as in Eq. 5. The key challenge lies in computing $E_s$, as the floating-point grid consists of non-uniform intervals, with spacing that doubles every $2^M$ points. We define $R$ such grid regions (subintervals) and denote each region by $\mathcal{R}_r$. The number of such regions is $R = 2\left\lfloor\frac{k_{max}}{2^M}\right\rfloor + 2\mathbb{I}(k_{max} \bmod 2^M > 0) - 3$ where $k_{max} = 2^{M+E} - c - 1$ is the index of the last representable grid point in the positive half-space.

Each region $\mathcal{R}_r$ corresponds to an interval:

$$\mathcal{R}_r = \begin{cases} [-s2^{\lfloor R/2\rfloor - r + 3 - b} + z, -s2^{\lfloor R/2\rfloor - r + 2 - b} + z], & r \leq \lfloor R/2\rfloor \\ [-s2^{2-b} - z, s2^{2-b} + z], & r = \lfloor R/2\rfloor + 1 \\ [s2^{r - \lfloor R/2\rfloor - b} + z, s2^{r - \lfloor R/2\rfloor + 1 - b} + z], & r > \lfloor R/2\rfloor + 1 \end{cases} \tag{11}$$

The quantization step size within region $r$ is denoted as $v_r = s2^{|r - 1 - \lfloor R/2\rfloor| + 1 - M - b}$.

Following the stochastic uniform error model, we model the stepping error in each region as conditionally uniform. Thus, the overall error distribution becomes a mixture of uniform distributions:

$$p(e_s) = \sum_{r=1}^{2R-1} \frac{1}{v_r}\mathbb{I}\left(-\frac{v_r}{2} \leq e \leq \frac{v_r}{2}\right) \cdot p(x \in \mathcal{R}_r), \tag{12}$$

where $p(x \in \mathcal{R}_r)$ denotes the probability mass of region $r$ under $p(x)$. (Appendix H gives an illustration of the stepping error). The expected stepping error is then:

$$E_s = \sum_{r=1}^{2R-1} \frac{v_r^2}{12} \cdot p(x \in \mathcal{R}_r). \tag{13}$$

### 3.2.1 Iterative Floating-Point Quantizer

To optimize the parameters $s$ and $z$, we employ an alternating optimization approach again. The scale update is $s = N_s(x)/D_s(x)$ where:

$$N_s(x) = \sum_{k=0}^{N/2-2} g_{-k+(N-3)/2} \int_{t_k}^{t_{k+1}} (x-z)p(x)dx + \sum_{k=N/2-1}^{N-2} g_{k-(N-3)/2} \int_{t_k}^{t_{k+1}} (x-z)p(x)dx, \quad (14)$$

$$D_s(x) = \sum_{k=0}^{N/2-2} g^2_{-k+(N-3)/2} \int_{t_k}^{t_{k+1}} p(x)dx + \sum_{k=N/2-1}^{N-2} g^2_{k-(N-3)/2} \int_{t_k}^{t_{k+1}} p(x)dx. \quad (15)$$

The shift $z$ is updated as $z = \frac{N_z(x)}{D_z(x)}$ with:

$$N_z(x) = \sum_{k=0}^{N/2-2} \int_{t_k}^{t_{k+1}} (x - sg_{-k+(N-3)/2})p(x)dx + \sum_{k=N/2-1}^{N-2} \int_{t_k}^{t_{k+1}} (x - sg_{k-(N-3)/2})p(x)dx, \quad (16)$$

$$D_z(x) = \sum_{k=0}^{N-2} \int_{t_k}^{t_{k+1}} p(x)dx. \quad (17)$$

where the denominator $D_z(x)$ sums up to the total number of points $I$. Then, we update the thresholds as $t_k = s(l_k + l_{k+1})/2 + z$ for $k = 1 : N-2$ where $t_0 = -\infty$ and $t_{N-1} = \infty$. In summary, in the first step, the data points are quantized given the previous values of $s$ and $z$, and in the second step, $s$ and $z$ are updated given the quantized data. This procedure resembles a constrained 1D $k$-means clustering, but where cluster centers must lie on a scaled floating-point grid.

### 3.2.2 Normal-Optimal Analytic Float Quantizer

Assuming $x \sim \mathcal{N}(0, \sigma^2)$, we can evaluate $p(x \in \mathcal{R}_r)$ analytically:

$$p(x \in \mathcal{R}_r) = F(\mathcal{R}_r^h) - F(\mathcal{R}_r^l), \quad (18)$$

where $F(x)$ is the Gaussian CDF and $\mathcal{R}_r^l, \mathcal{R}_r^h$ denote region bounds. From Eqs. 13 and 8, the total signal-to-noise ratio becomes:

$$SNR = \left[ 2\left(1 + \frac{C^2}{\sigma^2}\right) \mathcal{Q}\left(\frac{C}{\sigma}\right) - \frac{C}{\sigma}\sqrt{\frac{2}{\pi}} e^{-\frac{C^2}{2\sigma^2}} + \sum_{r=1}^{2R-1} \frac{v_r^2}{12\sigma^2}(F(\mathcal{R}_r^h) - F(\mathcal{R}_r^l)) \right]^{-1} \quad (19)$$

As in the uniform case, we fix $\sigma^2 = 1$ and numerically find $C_{opt}$ offline that maximizes SNR. We then compute scale as $s = C_{opt}\sigma_{(x)}/g_{2^{M+E}-c-1}$, and derive levels using Eq. 2 and Eq. 3.

### 3.3 Quantized Training Recipe

In quantization-aware training (QAT), quantization operations are inserted into the forward pass to simulate inference-time behavior, while gradients are propagated using the straight-through estimator (STE) (16) (Appendix C). However, fully optimizing quantization parameters at each training step is computationally prohibitive, particularly for iterative methods that require multiple passes to converge.

To address this, we adopt a single-step update scheme where the parameters at time step $t$, denoted $s^t$ and $z^t$, are updated using the values from the previous step ($s^{t-1}$, $z^{t-1}$) via a single iteration. This incremental update approximates convergence while maintaining training efficiency. The procedure is outlined in Algorithm 1 for iterative quantizers. In contrast, analytic methods do not require convergence, as the updates are in closed form. Algorithm 2 illustrates a single quantization step employed at time step $t$. See Appendix J for the initialization of the iterative quantizer, and Appendix **??** for the relative update speeds of the quantizers.

---

**Algorithm 1** Iterative Quantization Step

$Q_k(x \mid s, z)$ assigns level indices; $Q(x \mid s, z)$ returns dequantized values. See Eqs. (26)–(30).

---

1: **Input:** $\boldsymbol{x} \in \mathbb{R}^I$, previous scale $s^{t-1}$, shift $z^{t-1}$
2: Quantize: $Q_k(\boldsymbol{x} \mid s^{t-1}, z^{t-1})$
3: Update scale:

$$s^t = \frac{\sum_j (x_j - z^{t-1}) \cdot Q_k(x_j \mid s^{t-1}, z^{t-1})}{\sum_j Q_k(x_j \mid s^{t-1}, z^{t-1})^2}$$

4: Update shift:

$$z^t = \frac{1}{I} \sum_j x_j - s^{t-1} \cdot Q_k(x_j \mid s^{t-1}, z^{t-1})$$

5: Dequantize: $Q(\boldsymbol{x} \mid s^t, z^t)$
6: **Output:** $Q(\boldsymbol{x} \mid s^t, z^t), s^t, z^t$

---

**Algorithm 2** Analytic Quantization Step

Parameters computed via analytic method. $Q_k$ and $Q$ as defined in Eqs. (26)–(30).

---

1: **Input:** $\boldsymbol{x} \in \mathbb{R}^I$, optimal clipping point $C_{\text{opt}}$
2: Compute mean: $\mu_x = \frac{1}{I} \sum_j x_j$
3: Compute variance: $\sigma_x^2 = \frac{1}{I} \sum_j (x_j - \mu_x)^2$
4: Set shift: $z = \mu_x$
5: Set scale: $s = 2C_{\text{opt}} \cdot \sigma_x / (N-1)$
6: Quantize: $Q_k(\boldsymbol{x} \mid s, z)$
7: Dequantize: $Q(\boldsymbol{x} \mid s, z)$
8: **Output:** $Q(\boldsymbol{x} \mid s, z)$

---

# 4 Experiments

## 4.1 Signal-to-Noise Ratio (SNR) Analysis

We first present an error analysis of different data formats based on the error models derived in Section 3 for zero-mean, unit-variance Gaussian distributed data. This enables direct computation of the signal-to-noise ratio (SNR) using the analytic formulas in Eq. 10 and Eq. 19. Figure 1 illustrates the resulting SNR curves as a function of the clipping point for 4-bit floating-point and 4/3/2-bit uniform quantizers. Results indicate that the 4-bit uniform quantizer theoretically can outperform the E2M1 FP4 floating-point quantizer when Gaussian distribution data is quantized. However, a floating-point quantizer offers more robustness in case clipping points are suboptimal. On the other hand, Figure 1 shows that E2M1 FP4 format achieves higher peak SNR than E3M0, but E3M0 shows resilience to suboptimal clipping due to its larger dynamic range. This analysis suggests that E2M1 FP4 and INT4/3/2 are proper choices when weight-only quantization-aware training is performed under L2 regularization, which intrinsically forces normally distributed data.

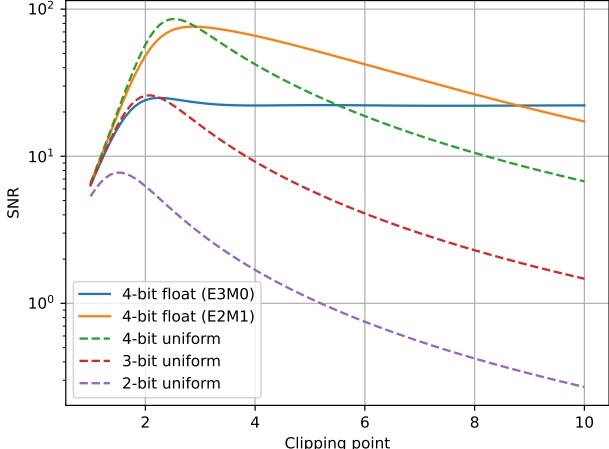

Figure 1: Signal-to-noise ratio versus clipping point for 4-bit float and 4/3/2-bit uniform quantizers, assuming zero-mean unit variance Gaussian input.

## 4.2 Quantization-Aware Training (QAT)

We experiment with end-to-end quantization-aware training, focusing on ResNet, MobileLLM, and Llama models. ResNet is used in our ablation study to analyze the effects of quantizer choice during training. We then evaluate

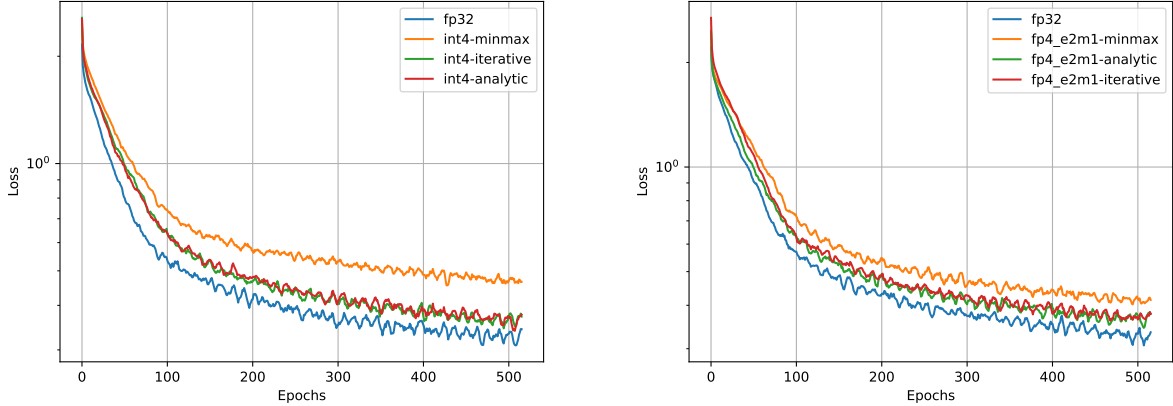

Figure 2: Loss functions with 4-bit uniform (left) and float (right) quantization across 40 epochs.

MobileLLM and Llama models to demonstrate that the proposed quantizers can achieve competitive or state-of-the-art accuracy. All experiments are conducted in 4-bit precision, representing the lowest precision currently supported by standardized floating-point formats.

### 4.2.1 Ablation Study with ResNet

For the ablation study, we use the ResNet-18 model trained on CIFAR-10 in a QAT framework where both activations and weights are quantized. The training recipe uses stochastic gradient descent with momentum $0.9$ and a weight decay coefficient of $5 \times 10^{-4}$.

The baseline training scheme uses FP32 precision. We compare the performance of min–max (FP4/INT4-minmax), analytic (FP4/INT4-analytic), and iterative quantization schemes (FP4/INT4-iterative). In the min–max training scheme, min–max quantizers are applied to both activations and weights in all quantizable layers, including linear and 2D convolution layers in ResNet. The iterative scheme uses iterative quantizers for both activations and weights. Because activation distributions are typically arbitrary, the FP4/INT4-analytic scheme uses iterative quantizers for activations and analytic quantizers for weights. The granularity is consistently tensor-wise across all experiments.

Figure 2.a shows training curves (log scale) over 40 epochs for uniform quantizers. Similarly, Figure 2.b shows training curves for floating-point quantizers. The loss curves consistently indicate inferior performance for the min–max quantizer (see Appendix F for further discussion). We observe that analytic and iterative quantization schemes perform similarly throughout training.

The final trained models achieve the following test losses: $[0.0063, 0.0057, 0.0058]$ for min–max, analytic, and iterative INT4 quantization, respectively, and $[0.0065, 0.0059, 0.0055]$ for the corresponding floating-point quantizers. The full-precision model achieves a test loss of $0.0050$. These results indicate that analytic and iterative quantizers can achieve similar performance, particularly in the INT4 setting. The performance of the analytic quantizer degrades slightly in FP4, which aligns with the SNR analysis presented in the previous section.

### 4.2.2 LLM performance

We compare StatQAT with state-of-the-art QAT frameworks, LLM-QAT (10) and ParetoQ (17), on MobileLLM (125M/600M) and Llama 3.2 (1B/3B/8B) models. Evaluation was conducted on zero-shot common-sense reasoning tasks and WikiText-2.

Each model was fine-tuned using AdamW with a learning rate of $2e{-}5$, L2 regularization of $0.01$, sequence length of 1024, and a batch size of 2 per GPU across 8 V100 GPUs using the WikiText-2 training split. Epoch durations range from under 30 seconds for the 125M model to approximately 38 minutes for the 8B model. To match baseline settings,

we apply symmetric weight-only quantization. For FP4, we use the E2M1 format based on the prior SNR analysis. Baseline metrics were collected using FP16 weights with FP32 accumulations due to hardware limitations.

Table 1 summarizes the performance of the proposed StatQAT quantizers compared with FP16 fine-tuning and existing QAT baselines. The iterative quantizer consistently matches or improves upon the performance of LLM-QAT and ParetoQ across model sizes and quantization configurations. The analytic quantizer achieves comparable performance to the iterative variant on smaller models but scales less effectively to larger models.

We observe that baseline algorithms degrade significantly under tensor-wise granularity, whereas StatQAT quantizers maintain performance more effectively. In the tensor-wise setting, a single clipping value must cover a wider dynamic range including outliers, making optimal clipping more critical. This phenomenon is well known and is consistent across all methods evaluated. StatQAT shows notable gains in this regime, indicating its ability to estimate effective clipping points.

In channel-wise granularity, the iterative StatQAT quantizer performs competitively with or better than LLM-QAT and ParetoQ across most configurations. Because each channel has its own clipping value, the dynamic range is narrower, and the optimization problem becomes easier, reducing the performance gap between methods. Nevertheless, for larger models, the StatQAT iterative quantizer consistently reduces the gap between FP16 and QAT baselines. While the improvements are modest, they were reproducible across multiple datasets, suggesting statistical significance.

## 5 Related Work

Quantization reduces memory, compute, and bandwidth requirements in deep neural networks by lowering numerical precision. Early PTQ methods relied on heuristic techniques such as min–max scaling or k-means clustering (18; 3), which often failed at low bit-widths. To overcome these limitations, quantization-aware training (QAT) was introduced, simulating quantization effects during training to improve robustness (4). Frameworks such as DoReFa-Net (7), PACT (8), and LSQ (19) proposed gradient-based updates for clipping and scaling, enabling strong accuracy retention in the 4-bit regime. More recent QAT methods—including LLM-QAT (10) and ParetoQ (17)—extend these ideas to large language models (LLMs).

From a theoretical standpoint, reducing mean-squared quantization error has motivated principled designs such as differentiable quantization objectives (20) and learned quantizer families (21; 22). However, these approaches typically rely on iterative optimization within each training step (23), which incurs additional computational overhead and becomes challenging for QAT on large models.

In parallel, LLM-scale models have driven the development of PTQ strategies tailored to transformer architectures. GPTQ (24) achieves high-fidelity INT4 quantization through Hessian-based error modeling, while AWQ (25) improves robustness via per-channel clipping and scale decoupling. SmoothQuant (26) redistributes activation magnitude into weights, achieving reliable INT8 quantization. These approaches highlight the importance of error compensation and outlier handling, but they operate primarily in the integer quantization domain.

More recently, FP8 and FP4 formats have emerged as promising alternatives for both training and inference (27; 28). Kuzmin et al. (9) treat clipping as a trainable parameter, similar to LSQ, while Liu et al. (10) optimize FP4 clipping using online search in a PTQ setting. However, these methods rely on iterative search procedures, which can be impractical for QAT where quantization parameters must be updated at every training step.

Although methods such as LSQ and PACT also update clipping or scale during QAT, they do so via stochastic gradient-based optimization and do not exploit any closed-form structure of the quantization error. Classical quantizer analyses are primarily designed for unconstrained non-uniform quantizers and do not directly apply to the hardware-restricted uniform and FP4 formats increasingly supported by modern accelerators. Consequently, prior work provides limited analytic guidance for the highly constrained layouts of FP4 formats such as E2M1 and E3M0.

Statistics-aware methods have been explored for integer quantization; for example, SAWB (29) selects clipping points under Gaussian assumptions. However, such approaches rely on the uniform and symmetric structure of integer quantizers. In contrast, FP4 quantizers exhibit non-uniform, magnitude-dependent spacing, asymmetric representable sets, and tight coupling between scale, clipping, and exponent allocation, fundamentally altering the quantization error structure. These constraints make existing INT-focused analytical approaches difficult to apply directly.

| Model | Type | Granularity | Method | ARC-e | ARC-c | BoolQ | PIQA | SIQA | HellaSwag | OBQA | WinoGrande | Avg. | Wiki2 |
|---|---|---|---|---|---|---|---|---|---|---|---|---|---|
| MobileLLM 125M | | | FP16 Baseline | 46.0 | 20.3 | 57.8 | 64.7 | 38.2 | 32.7 | 18.0 | 52.5 | 41.3 | 10.3 |
| | FP4 | Tensor | LLM-QAT | 38.3 | 19.5 | 43.6 | 56.8 | 35.9 | 30.0 | 13.0 | 50.3 | 35.9 | 15.6 |
| | | | ParetoQ | 38.5 | 18.9 | 56.2 | 58.9 | 35.4 | 30.1 | 15.0 | 50.4 | 37.9 | 15.4 |
| | | | StatQAT-iterative | 44.6 | 20.1 | 54.3 | 61.3 | 37.8 | 31.7 | 18.0 | 53.1 | **40.1** | **11.4** |
| | | | StatQAT-analytic | 43.7 | 19.1 | 55.1 | 61.6 | 37.4 | 31.8 | 16.2 | 52.0 | 39.6 | 11.6 |
| | | Channel | LLM-QAT | 44.7 | 21.0 | 55.9 | 63.7 | 37.5 | 32.3 | 18.4 | 50.8 | 40.5 | **11.0** |
| | | | ParetoQ | 44.1 | 20.9 | 55.1 | 63.4 | 37.3 | 32.3 | 18.8 | 50.7 | 40.3 | **11.0** |
| | | | StatQAT-iterative | 44.9 | 20.1 | 57.2 | 63.2 | 37.4 | 32.1 | 19.0 | 51.1 | 40.6 | **10.9** |
| | | | StatQAT-analytic | 45.4 | 21.8 | 60.5 | 61.8 | 37.6 | 32.1 | 17.0 | 51.9 | **41.0** | 11.2 |
| | INT4 | Tensor | LLM-QAT | 29.8 | 19.4 | 49.4 | 53.5 | 33.7 | 27.3 | 13.6 | 50.4 | 34.6 | 32.4 |
| | | | ParetoQ | 30.3 | 19.7 | 48.4 | 55.6 | 34.3 | 27.5 | 10.8 | 50.7 | 34.7 | 31.3 |
| | | | StatQAT-iterative | 43.9 | 20.5 | 53.7 | 61.8 | 36.7 | 31.6 | 17.0 | 53.1 | **39.8** | **11.8** |
| | | | StatQAT-analytic | 42.7 | 20.6 | 55.1 | 61.9 | 37.2 | 31.9 | 16.2 | 52.2 | **39.7** | **11.8** |
| | | Channel | LLM-QAT | 44.7 | 20.1 | 56.4 | 62.7 | 38.2 | 32.2 | 17.4 | 52.2 | **40.5** | 11.3 |
| | | | ParetoQ | 44.4 | 19.5 | 58.1 | 62.8 | 37.9 | 32.0 | 16.4 | 52.3 | 40.4 | 11.4 |
| | | | StatQAT-iterative | 43.3 | 19.8 | 51.5 | 62.9 | 37.6 | 32.2 | 16.8 | 53.0 | 39.6 | **11.1** |
| | | | StatQAT-analytic | 45.5 | 21.2 | 55.6 | 62.7 | 36.7 | 32.0 | 19.0 | 51.2 | **40.5** | 11.3 |
| MobileLLM 600M | | | FP16 Baseline | 60.6 | 26.7 | 63.6 | 70.0 | 41.0 | 43.0 | 21.8 | 60.7 | 48.4 | 7.4 |
| | FP4 | Tensor | LLM-QAT | 46.4 | 21.7 | 49.5 | 61.9 | 37.2 | 35.5 | 16.0 | 52.9 | 40.1 | 11.1 |
| | | | ParetoQ | 46.3 | 22.8 | 48.9 | 62.4 | 37.5 | 36.0 | 15.6 | 52.8 | 40.3 | 11.0 |
| | | | StatQAT-iterative | 58.9 | 27.1 | 62.1 | 69.5 | 38.5 | 41.2 | 21.8 | 59.1 | **47.3** | **8.0** |
| | | | StatQAT-analytic | 57.1 | 27.6 | 62.2 | 69.4 | 38.7 | 40.0 | 21.4 | 56.4 | 46.6 | 8.5 |
| | | Channel | LLM-QAT | 58.9 | 25.7 | 62.9 | 70.1 | 40.3 | 42.0 | 22.2 | 57.9 | **47.5** | **7.7** |
| | | | ParetoQ | 58.9 | 25.7 | 63.1 | 69.5 | 39.9 | 41.9 | 21.2 | 58.0 | 47.3 | **7.7** |
| | | | StatQAT-iterative | 59.3 | 26.5 | 62.3 | 68.7 | 39.3 | 42.2 | 22.0 | 60.5 | **47.6** | **7.7** |
| | | | StatQAT-analytic | 59.5 | 27.0 | 62.6 | 69.4 | 38.3 | 40.9 | 20.8 | 58.5 | 47.1 | 8.1 |
| | INT4 | Tensor | LLM-QAT | 29.4 | 20.2 | 45.5 | 53.6 | 34.0 | 27.6 | 13.6 | 50.4 | 34.3 | 32.8 |
| | | | ParetoQ | 29.5 | 19.2 | 39.9 | 53.3 | 33.6 | 27.6 | 13.2 | 50.2 | 33.3 | 32.2 |
| | | | StatQAT-iterative | 57.5 | 26.6 | 62.2 | 68.3 | 37.5 | 40.2 | 21.8 | 58.0 | **46.5** | **8.3** |
| | | | StatQAT-analytic | 56.5 | 25.0 | 60.6 | 68.6 | 38.2 | 39.7 | 20.4 | 57.1 | 45.8 | 8.7 |
| | | Channel | LLM-QAT | 58.4 | 25.6 | 62.8 | 69.3 | 39.5 | 41.5 | 21.4 | 58.7 | 47.2 | 7.9 |
| | | | ParetoQ | 58.7 | 25.4 | 63.5 | 69.6 | 38.9 | 41.7 | 21.4 | 58.6 | 47.2 | 7.9 |
| | | | StatQAT-iterative | 59.6 | 26.4 | 63.1 | 69.2 | 39.4 | 41.8 | 21.8 | 58.9 | **47.5** | **7.8** |
| | | | StatQAT-analytic | 57.4 | 26.1 | 62.6 | 68.5 | 38.3 | 40.5 | 22.8 | 57.8 | 46.8 | 8.2 |
| Llama 3.2 1B | | | FP16 Baseline | 63.8 | 30.9 | 61.9 | 73.0 | 39.8 | 44.1 | 25.2 | 58.0 | 49.6 | 10.5 |
| | FP4 | Tensor | LLM-QAT | 27.2 | 20.2 | 37.8 | 51.3 | 34.2 | 25.7 | 12.2 | 50.4 | 32.4 | 487.0 |
| | | | ParetoQ | 26.0 | 20.5 | 37.8 | 50.9 | 34.4 | 25.9 | 11.0 | 49.2 | 32.0 | 530.3 |
| | | | StatQAT-iterative | 58.6 | 28.3 | 64.2 | 70.5 | 39.6 | 42.4 | 23.4 | 57.9 | **48.1** | **11.2** |
| | | | StatQAT-analytic | 58.6 | 29.4 | 62.1 | 70.0 | 41.9 | 41.7 | 22.8 | 56.0 | 47.8 | 11.5 |
| | | Channel | LLM-QAT | 59.8 | 28.5 | 59.4 | 70.8 | 39.8 | 41.8 | 23.6 | 57.2 | 47.6 | 11.4 |
| | | | ParetoQ | 61.2 | 29.2 | 59.8 | 70.9 | 39.3 | 41.9 | 21.6 | 55.9 | 47.5 | 11.4 |
| | | | StatQAT-iterative | 60.7 | 29.6 | 51.7 | 72.0 | 40.1 | 42.5 | 21.8 | 58.4 | 47.1 | **10.9** |
| | | | StatQAT-analytic | 61.4 | 29.7 | 61.4 | 70.8 | 40.4 | 41.8 | 26.2 | 55.6 | **48.4** | 11.2 |
| | INT4 | Tensor | LLM-QAT | 25.9 | 21.0 | 37.8 | 52.3 | 34.5 | 25.9 | 12.6 | 50.0 | 32.5 | 1727.9 |
| | | | ParetoQ | 26.6 | 20.2 | 37.8 | 52.1 | 34.5 | 25.9 | 12.0 | 47.8 | 32.1 | 1752.6 |
| | | | StatQAT-iterative | 58.6 | 28.1 | 61.7 | 70.3 | 40.1 | 41.4 | 22.0 | 54.5 | **47.1** | **11.6** |
| | | | StatQAT-analytic | 54.9 | 26.4 | 59.1 | 70.4 | 39.5 | 40.3 | 21.6 | 54.6 | 45.9 | 12.0 |
| | | Channel | LLM-QAT | 60.9 | 29.7 | 63.4 | 70.9 | 39.0 | 40.5 | 22.0 | 54.9 | **47.7** | 12.1 |
| | | | ParetoQ | 61.5 | 28.1 | 61.6 | 70.3 | 38.8 | 39.8 | 22.0 | 56.0 | 47.3 | 12.2 |
| | | | StatQAT-iterative | 60.9 | 29.1 | 59.2 | 71.9 | 38.8 | 41.8 | 23.2 | 56.0 | **47.6** | **11.1** |
| | | | StatQAT-analytic | 59.0 | 29.9 | 58.7 | 70.5 | 39.5 | 41.1 | 22.6 | 57.2 | 47.3 | 11.4 |
| Llama 3.2 3B | | | FP16 Baseline | 73.7 | 43.0 | 73.0 | 75.5 | 41.5 | 49.9 | 30.0 | 66.9 | 56.7 | 8.8 |
| | FP4 | Tensor | LLM-QAT | 26.1 | 21.7 | 37.8 | 52.3 | 34.0 | 25.9 | 9.6 | 49.1 | 32.1 | 2331.9 |
| | | | ParetoQ | 25.7 | 21.8 | 37.8 | 51.4 | 35.0 | 25.8 | 12.4 | 51.5 | 32.7 | 1907.7 |
| | | | StatQAT-iterative | 69.7 | 38.7 | 69.9 | 73.0 | 41.8 | 47.1 | 26.4 | 61.6 | **53.5** | **9.6** |
| | | | StatQAT-analytic | 43.7 | 22.8 | 62.2 | 60.6 | 36.3 | 32.4 | 15.0 | 52.8 | 40.7 | 17.8 |
| | | Channel | LLM-QAT | 73.9 | 41.1 | 72.9 | 74.6 | 40.4 | 48.9 | 29.8 | 66.8 | **56.1** | 9.3 |
| | | | ParetoQ | 73.5 | 41.8 | 73.6 | 75.1 | 40.1 | 48.9 | 29.2 | 66.5 | **56.1** | 9.4 |
| | | | StatQAT-iterative | 71.7 | 39.9 | 72.0 | 74.4 | 41.4 | 48.9 | 28.8 | 65.7 | 55.4 | **9.0** |
| | | | StatQAT-analytic | 43.1 | 24.4 | 62.4 | 61.7 | 37.5 | 33.8 | 15.4 | 52.8 | 41.4 | 17.3 |
| | INT4 | Tensor | LLM-QAT | 25.9 | 21.2 | 37.8 | 51.3 | 35.0 | 26.1 | 12.8 | 48.5 | 32.3 | 2084.8 |
| | | | ParetoQ | 26.4 | 20.5 | 37.8 | 52.2 | 34.3 | 26.0 | 13.2 | 48.9 | 32.4 | 1562.7 |
| | | | StatQAT-iterative | 63.8 | 34.7 | 64.1 | 71.3 | 39.3 | 44.0 | 22.0 | 59.8 | **49.9** | **10.0** |
| | | | StatQAT-analytic | 44.9 | 22.8 | 62.2 | 61.6 | 36.4 | 33.0 | 17.0 | 54.0 | 41.5 | 17.2 |
| | | Channel | LLM-QAT | 70.2 | 38.3 | 70.9 | 73.7 | 41.0 | 47.1 | 27.4 | 64.6 | 54.1 | 9.6 |
| | | | ParetoQ | 71.2 | 39.1 | 71.4 | 73.3 | 41.0 | 47.0 | 28.4 | 65.0 | 54.5 | 9.5 |
| | | | StatQAT-iterative | 70.3 | 38.9 | 72.3 | 74.4 | 40.3 | 48.4 | 29.2 | 64.4 | **54.8** | **9.2** |
| | | | StatQAT-analytic | 52.1 | 25.3 | 62.8 | 63.6 | 37.5 | 36.3 | 17.2 | 53.7 | 43.6 | 13.5 |
| Llama 3 8B | | | FP16 Baseline | 78.8 | 47.5 | 79.0 | 78.2 | 42.6 | 55.2 | 32.2 | 72.7 | 60.8 | 7.6 |
| | FP4 | Tensor | LLM-QAT | 26.6 | 20.7 | 37.8 | 51.1 | 34.4 | 25.9 | 12.0 | 49.9 | 32.3 | 2562.4 |
| | | | ParetoQ | 26.9 | 20.2 | 37.8 | 50.4 | 34.4 | 25.8 | 12.4 | 48.1 | 32.0 | 2473.2 |
| | | | StatQAT-iterative | 74.2 | 42.3 | 77.9 | 75.4 | 43.7 | 54.5 | 29.2 | 68.4 | **58.2** | **8.1** |
| | | | StatQAT-analytic | 31.8 | 20.0 | 59.4 | 54.4 | 35.6 | 26.9 | 13.4 | 49.6 | 36.4 | 75.2 |
| | | Channel | LLM-QAT | 77.0 | 45.4 | 75.8 | 77.3 | 42.2 | 52.7 | 32.8 | 72.7 | 59.5 | 8.2 |
| | | | ParetoQ | 77.5 | 46.2 | 79.7 | 77.7 | 42.4 | 52.4 | 32.4 | 72.8 | **60.1** | 8.1 |
| | | | StatQAT-iterative | 76.1 | 45.1 | 78.8 | 77.0 | 43.3 | 54.1 | 31.8 | 71.3 | 59.7 | **7.9** |
| | | | StatQAT-analytic | 48.1 | 24.7 | 68.7 | 61.4 | 38.0 | 40.1 | 19.0 | 52.4 | 44.0 | 12.2 |
| | INT4 | Tensor | LLM-QAT | 26.2 | 19.6 | 37.8 | 51.1 | 34.6 | 26.0 | 11.6 | 49.1 | 32.0 | 2388.0 |
| | | | ParetoQ | 26.5 | 20.4 | 37.8 | 51.0 | 34.6 | 26.0 | 11.2 | 49.3 | 32.2 | 2444.8 |
| | | | StatQAT-iterative | 68.8 | 35.8 | 75.4 | 71.9 | 42.2 | 49.8 | 25.4 | 65.6 | **54.4** | **9.2** |
| | | | StatQAT-analytic | 25.8 | 20.4 | 37.8 | 50.8 | 34.9 | 25.9 | 10.4 | 50.1 | 32.0 | 732.3 |
| | | Channel | LLM-QAT | 75.9 | 45.4 | 75.3 | 77.0 | 42.6 | 52.4 | 29.2 | 69.2 | 58.4 | 8.5 |
| | | | ParetoQ | 75.2 | 44.0 | 76.0 | 76.9 | 42.6 | 52.5 | 30.2 | 71.1 | 58.6 | 8.5 |
| | | | StatQAT-iterative | 76.6 | 45.5 | 79.7 | 77.3 | 43.1 | 53.6 | 33.4 | 72.1 | **60.2** | **8.0** |
| | | | StatQAT-analytic | 32.2 | 22.0 | 53.3 | 55.2 | 35.8 | 28.0 | 14.0 | 53.4 | 36.7 | 81.8 |

Table 1: Accuracy comparison of proposed StatQAT quantizers with baselines on benchmark datasets. Our test setup included the implementation of the baselines for a fair comparison. ±0.1 variance is accounted for when bolding the best performance for statistical significance.

Our work differs from prior methods by providing a unified analytic error formulation for both uniform and floating-point quantizers, together with closed-form updates suitable for QAT. The proposed analytic and iterative quantizers avoid expensive iterative search, incur negligible overhead, and achieve accuracy competitive with or exceeding state-of-the-art iterative methods, including LSQ and ParetoQ, in our LLM evaluations. To the best of our knowledge, this work represents one of the first statistics-aware quantization frameworks providing closed-form updates for FP4 formats within QAT, offering both theoretical insight and practical relevance for large-scale training.

## 6 Conclusion

We presented a statistical framework for analyzing and minimizing quantization error in uniform and floating-point quantization schemes. Building on this framework, we proposed iterative and analytic quantizers effective for both activations and weights. Our methods enable accurate, low-overhead quantization-aware training by providing efficient estimation of quantization parameters during training. Our error analysis and experiments on ResNet and LLMs demonstrate that the proposed quantizers achieve competitive or state-of-the-art performance across several low-precision settings. Notably, the analytic quantizers achieve performance comparable to the iterative variants at a fraction of the computational cost, making them particularly practical for training.

These results highlight the potential of statistical error modeling as a principled approach to quantizer design. While our study focuses on 4-bit quantization due to current hardware support for formats such as FP4, extending this framework to ultra-low precision and more complex data distributions remains an important direction for future research.

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

## A  Practical Implementation of Uniform Quantization

In real-world applications, instead of storing high-precision $l_k$ values corresponding to each data point, only the indexes are stored in integer format for reduced data size. Updating Eq.1 to return indexes instead of levels as

$$Q_k(x|\boldsymbol{l}, \boldsymbol{t}) = k, \quad \text{if } t_k < x \leq t_{k+1}, \tag{20}$$

we obtain the unsigned integer representation of point $x$. For signed integer representation, the function returns $k - N/2 + 1$.

We can alternatively compute the index for uniform unsigned integer quantization as follows:

$$Q_k(x|s, z) = clip(\lfloor \frac{x - z}{s} \rceil, n, m) \tag{21}$$

where $\lfloor \rceil$ is the integer rounding, and $clip(x, n, m)$ is the clipping operator where $n, m$ defines the boundaries such that $n = 0, m = N - 2$. In this case, the level is computed as:

$$Q(x|s, z) = s\hat{k} + z \tag{22}$$

where $\hat{k} = Q_k(x|s, z)$ is the quantization index of $x$.

For signed integer quantization, the index is computed as follows

$$Q_k(x|s, z) = clip(\lfloor \frac{x - z}{s} \rceil, n, m) - \frac{N}{2} + 1 \tag{23}$$

and the levels as $Q(x|s, z) = s(\hat{k} + N/2 - 1) + z$.

## B  Practical Implementation of Float Quantization

Given a higher precision floating point number $x$, we can quantize it to a lower precision floating point number by first computing the step size $v$ given the scaled and shifted input value:

$$v = \begin{cases} 2^{\lfloor \log_2 |\frac{x-z}{s}| \rfloor - M}, & \text{if } \lfloor \log_2 |\frac{x - z}{s}| + b \rfloor \geq 1 \\ 2^{1-M-b}, & \text{otherwise} \end{cases}$$

Then, we quantize $x$ with this step size

$$Q_k(x|s, z) = v * \text{clip}(\lfloor \frac{x - z}{sv} \rceil, n, m) \tag{24}$$

When we scale-shift back, we obtain dequantized data:

$$Q(x|s, z) = sQ_k(x|s, z) + z \tag{25}$$

where $m = -n = 2^{2^E - b - 2}(2 - 2^{-M})$. Note the input dependency of the step size due to the non-uniform normal grid. $Q_k(x|s, z)$ is the grid point in lower precision, which we store for reduced data size or low precision computation.

## C  Quantized Training

The training objective is to minimize the negative log-likelihood and a regularization term:

$$\mathcal{L}(\boldsymbol{\theta}) = -\log p(\boldsymbol{\theta}) - \sum_{n=1}^{N} \log p(\boldsymbol{y}_n | \boldsymbol{x}_n, \boldsymbol{\theta}), \tag{26}$$

where $\boldsymbol{\theta} \in \mathbb{R}^M$ is the set of model parameters. The prior is often factorized across layers: $\log p(\boldsymbol{\theta}) = \sum_{l=1}^{L} \log p(\boldsymbol{\theta}_l)$. Per-layer quantization can then be enforced via an additonal regularization term:

$$\mathcal{L}_r(\boldsymbol{\theta_l}) = \inf_{\boldsymbol{\theta}_0 \in l} ||\boldsymbol{\theta_l} - \boldsymbol{\theta}_0||_p = ||\boldsymbol{\theta_l} - Q(\boldsymbol{\theta_l}|l)||_p \tag{27}$$

which penalizes the $p$-norm difference between the original parameters and their quantized counterparts. Since $Q(\cdot)$ is non-differentiable (30), the straight-through estimator (STE) (16) is commonly used to approximate gradients of the regularization term. Particularly, the gradients are computed at the quantized parameters with the loss function in Eq. 26. Subsequently, the parameter updates are performed in unconstrained space.

Additionally, activations can be quantized in quantization-aware training for low precision deployment (19; 8). A linear layer using quantized weights $\boldsymbol{\theta}$ and activations $\boldsymbol{x}$ computes $\boldsymbol{\theta}^T x \approx Q(\boldsymbol{\theta}|l_{\boldsymbol{\theta}})Q(\boldsymbol{x}|l_{\boldsymbol{x}})$.

## D  Derivation of stepping error

$$E_s = \mathbb{E}[e_s^2] = \int e_s^2 p(e_s) de_s = \frac{1}{s} \int_{-\frac{s}{2}}^{\frac{s}{2}} e_s^2 de_s = \frac{s^2}{12}. \tag{28}$$

## E  Derivation of Clipping Error Function

The clipping error is defined as

$$N_c = 2 \int_C^\infty (x - C)^2 p(x) dx \tag{29}$$

where

$$p(x) = \frac{1}{\sigma\sqrt{2\pi}} e^{-\frac{x^2}{2\sigma^2}} \tag{30}$$

is the centered normal data distribution with $\sigma^2$ variance. We can decompose the error function as follows:

$$N_c = 2 \int_C^\infty x^2 p(x) dx + 2C^2 \int_C^\infty p(x) dx - 4C \int_C^\infty x p(x) dx \tag{31}$$

The integral in the first term is computed by integration by parts:

$$\begin{aligned}
\int_C^\infty x^2 p(x) dx &= \frac{1}{\sigma\sqrt{2\pi}} \int_C x^2 e^{-\frac{x^2}{2\sigma^2}} dx \\
&= \frac{\sigma^2}{2} erf\left(\sqrt{\frac{x}{2\sigma^2}}\right) - \frac{\sigma}{\sqrt{2\pi}} x e^{-\frac{x^2}{2\sigma^2}} \Big|_{x=C}^{x=\infty} \\
&= \sigma^2\left(\frac{1}{2} - \frac{1}{2} erf\left(\frac{C}{\sigma\sqrt{2}}\right)\right) + \frac{C\sigma}{\sqrt{2\pi}} e^{-\frac{C^2}{2\sigma^2}} \\
&= \sigma^2 Q\left(\frac{C}{\sigma}\right) + \frac{C\sigma}{\sqrt{2\pi}} e^{-\frac{C^2}{2\sigma^2}}
\end{aligned} \tag{32}$$

The integral in the second term is computed by the definition of the erf function:

$$\begin{aligned}
\int_C^\infty p(x) dx &= \frac{1}{\sigma\sqrt{2\pi}} \int_C^\infty e^{-\frac{x^2}{2\sigma^2}} dx \\
&= \frac{1}{2} erf\left(\sqrt{\frac{1}{2\sigma^2}} x\right) \Big|_{x=C}^{x=\infty} \\
&= \frac{1}{2}\left(1 - erf\left(\sqrt{\frac{1}{2\sigma^2}} x\right)\right) \\
&= Q\left(\frac{C}{\sigma}\right)
\end{aligned} \tag{33}$$

The integral in the third term is computed by integration by parts:

$$\int_C^\infty xp(x)dx = \frac{1}{\sigma\sqrt{2\pi}}\int_C^\infty xe^{-\frac{x^2}{2\sigma^2}}dx$$
$$= -\frac{C\sigma}{\sqrt{2\pi}}e^{-\frac{x^2}{2\sigma^2}}\Big|_{x=C}^{x=\infty} \tag{34}$$
$$= \frac{C\sigma}{\sqrt{2\pi}}e^{-\frac{C^2}{2\sigma^2}}$$

Plugging Eq. 32, Eq. 33, and Eq. 34 into Eq. 31 results in

$$N_c = 2(\sigma^2 + C^2)\mathcal{Q}(\frac{C}{\sigma}) - 2C\frac{\sigma}{\sqrt{2\pi}}e^{-\frac{C^2}{2\sigma^2}} \tag{35}$$

## F  Relation to Min-max Quantizer and Casting

### F.1  Uniform Quantization

Assume the data is uniformly distributed, $p(x) = Unif(x| - \alpha, \alpha)$. The clipping error is then computed analytically as follows:

$$E_c = 2\int_C^\infty (x - C)^2 p(x)dx = \frac{1}{\alpha}\int_C^\infty (x - C)^2 dx$$
$$= -\frac{C^3}{3\alpha} + C^2 - C\alpha + \frac{\alpha^2}{3} \tag{36}$$

while $E_s$ is the same as in Eq. 28. It can be observed that $E_c$ decreases cubically with $C$ until $\alpha$, while $E_s$ increases quadratically. Therefore, in the case of uniformly distributed data, the minimum error is achieved when $C$ is set to $\alpha$, resulting in zero clipping error. This indicates that the min-max quantizer is the optimal uniform quantizer in terms of mean squared quantizer error only when the data is uniformly distributed, which is not the case in deep learning model parameters or activations. Therefore, while this quantizer is commonly used in quantization tools, it has also been the primary source of quantization error due to high stepping error, especially in low-bit settings.

### F.2  Float Quantization

Directly casting the input data corresponds to using quantization levels without scaling and shifting. This method introduces large quantization errors if the data values are much smaller than the largest grid point (stepping error) or if the data values are larger than the largest grid (clipping error). These errors become more significant when we use lower bit quantization, such as in FP4 data types.

To avoid clipping errors, we can shift and scale the levels so that they match the data range using min-max quantization with $s = \frac{\alpha-\beta}{2l_{max}}$ and $z = \beta + sl_{max}$ where $l_{max}$ is the largest grid point floating data type can represent, $\alpha = \max(\boldsymbol{X})$ and $\beta = \min(\boldsymbol{X})$. Then we can transform the data as $(x - z)/s$ and do the typical float casting. This method zeros out the clipping error but introduces a large stepping error when there are outliers in the dataset. Unlike uniform quantization, the min-max float quantizer is not optimal for uniform distributed data. Indeed, there is no such tractable data distribution that this quantizer is optimal for due to base2 grid spacing.

## G  Optimization of non-uniform quantizers

Mean-squared quantization error is formulated as follows:

$$\mathbb{E}[e^2] = \mathbb{E}[(x - Q(x|\boldsymbol{l}, \boldsymbol{t}))^2] = \sum_{k=0}^{N-2}\int_{t_k}^{t_{t+1}}(x - l_k)^2 p(x)dx \tag{37}$$

where the errors corresponding to each quantization level are integrated over $N - 1$ intervals given the data distribution $p(x)$. The objective is to determine the quantization levels $\boldsymbol{l}$ and thresholds $\boldsymbol{t}$ to minimize the error function.

## G.1 Iterative Method

In non-uniform quantization, there are no constraints on the parameters. This allows for iterative optimization of the objective. We present an alternating optimization algorithm for this purpose. By taking the derivative of the quantization error in Eq. 37 with respect to $l_k$ and setting it to zero, we can derive the following update for the levels:

$$l_k = \mathbb{E}[x|t_k < x \le t_{k+1}] \tag{38}$$

for $k = 0 : N - 2$. This expectation is computed using the empirical mean of the data points in the interval. Taking the derivative with respect to $t_k$ yields the following update for the thresholds:

$$t_k = \frac{1}{2}(l_k + l_{k-1}) \tag{39}$$

for $k = 1 : N - 2$. Then, $t_0 = -\infty$, and $t_N = \infty$ are set to cover the unbounded data space. In summary, in the first step of the algorithm, input data is compared to the thresholds determined at the previous iteration and quantized accordingly, and in the second step, the levels and thresholds are updated given the recently quantized data. This algorithm converges to optimal mean-squared quantization error regardless of the data distribution when initialized properly. One can recognize that the iterative solution is a 1-dimensional k-means clustering solution, which has been used in (2) for deep learning model quantization.

## G.2 Normal-optimal Analytic Quantizer

The deep learning models have approximately normally distributed weights (11). Based on this assumption, we propose an analytic quantizer, which is fast and as accurate as the iterative method as long as the distribution assumption holds. The update equation 38 in the iterative method relies on computing the expectation of the data given an interval. Given the data is zero-mean normally distributed with unit variance, this expectation can be computed analytically without any samples via the first moment of the two-sided truncated normal distribution:

$$\mathbb{E}[x|t_k < x \le t_{k+1}] = -\frac{p(t_k + 1) - p(t_k)}{\mathcal{F}(t_{k+1}) - \mathcal{F}(t_k)} \tag{40}$$

Iterating through Eq. 40 and Eq. 39 converges to the optimal non-uniform quantizer for data with a centered unit variance normal distribution, in terms of mean squared quantization error. Unlike the iterative method, this optimization is performed offline without any data, and the computed optimal levels $l_{opt}$ and thresholds $t_{opt}$ are used afterward for all layers during the quantization process as follows: the mean and variance of each quantizable weight tensor are computed, and the quantizer levels are updated accordingly as $l = \sigma_{(x)} l_{opt} + m_{(x)}$, where $\sigma_{(x)}$ and $m_{(x)}$ are the empirical standard deviation and mean, respectively. This makes the online quantization process much faster, as only basic statistics are computed at that stage.

## G.3 Relation to Quantile Quantization

Another non-iterative method for non-uniform quantization is quantile quantization, which computes the quantiles of $N$ equally spaced probabilities and uses them as quantization levels (11). To find the quantile function, we first compute empirical CDF, then use its inverse to determine threshold levels as follows:

$$t_k = \mathcal{F}_x^{-1}(k/(N-1)) \tag{41}$$

for $k = 1 : N - 2$, $t_0 = -\infty$ and $t_{N-1} = \infty$. Then we estimate levels as $l_k = \mathbb{E}[x|t_k < x < t_{k+1}]$ for $k = 0 : N - 2$. In practice, we sort the input data points and assign the data points with indexes $k(I - 1)/(N - 1)$ as thresholds, where $I$ is the total number of data points. One can notice that if we use only levels to quantize by finding the closest levels using squared distance, we don't exactly equalize the histogram. Ideally, one should use thresholds to determine the index and then assign the level accordingly by Eq. 1 for improved equalization. Empirically, using levels with squared distance works better in terms of mean squared quantization error. Equalizing the bins does not necessarily result in lower quantization error. Indeed, this is true only for uniform distributed data, which is not the case for deep learning model weights.

### G.4   Comparison of quantization levels

Non-uniform quantizers have no constraints on choosing quantization levels. Here we show how non-uniform quantizers compare to each other, given a simulated input tensor of shape [256x256] from a zero-mean unit variance normal distribution. As seen in Figure 3, iterative and analytic quantizers converge to the same points for the normally distributed data. The quantizer errors are $MSE = [0.21, 0.21, 0.25]$, and $MAE = [0.41, 0.41, 0.41]$ for iterative, analytic, and quantile quantization, respectively. Quantile quantizer has a higher MSE than others, although all quantizers give a comparable MAE.

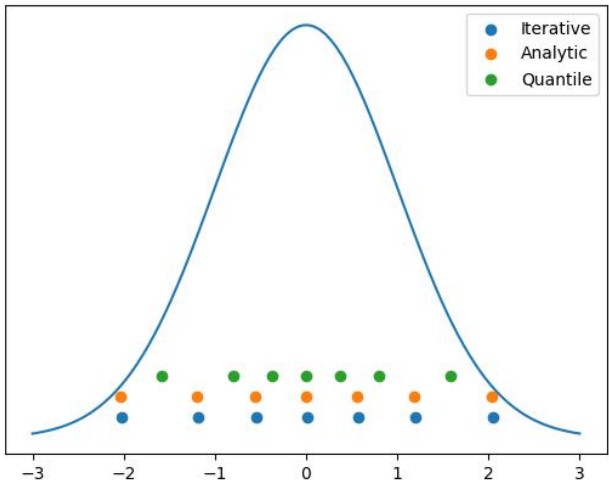

Figure 3: Inferred quantization levels of 3-bit non-uniform quantizers for a normally distributed input data.

## H   Stepping Error Distribution

We define total quantization error as the sum of clipping and stepping errors. Clipping error is obtained with quadratic error integrated over the data distribution. However, the stepping error depends on the quantization scheme. Figure 4 illustrates the differences in the stepping error distribution of uniform and float quantization schemes given a set of normally distributed input data with zero mean and unit variance. Regardless of the data distribution, quantizer type, and total bits, we obtain uniform distributed stepping error in a uniform quantization scheme. The error is bounded in $[-s/2, s/2]$ with probability $1/s$ as shown in Figure 4.a-c. From this observation, we compute the stepping error power as in Eq. 28. On the other hand, the error distribution is a mixture of uniforms in floating point quantization schemes due to the exponential spacing of every $2^M$ point. Thus, the error space widens at each consecutive region as shown in Figure 4.b-d. Hence, the stepping error power is computed by Eq. 13 since now the data distribution affects the mixture coefficient, which changes the stepping error power.

## I   SNR per Layer

Figure 5 reports the per-layer SNR of the ordered linear layers in Llama-3-1B under both the Analytic and MinMax quantization schemes. Across all layers, the Analytic method consistently achieves higher SNR, demonstrating its ability to more effectively reduce quantization error. Because this reduction compounds across forward passes, the Analytic approach yields more stable training dynamics and higher end-to-end accuracy relative to MinMax.

## J   Initialization of StatQAT-iterative

Quantizer initialization is critical to avoid early-stage gradient instabilities. For weights, if the initialization follows a normal distribution, as is typical under common initialization schemes and regularization, the analytic quantizer derived

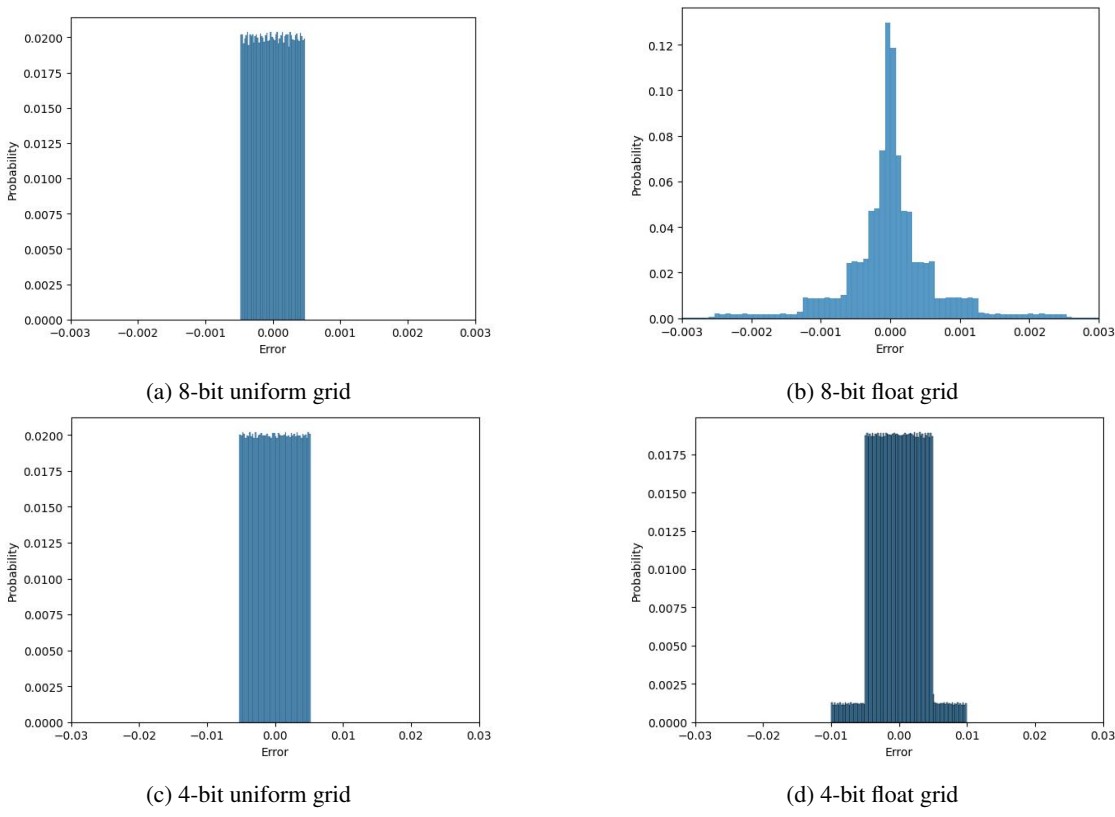

(a) 8-bit uniform grid

(b) 8-bit float grid

(c) 4-bit uniform grid

(d) 4-bit float grid

Figure 4: Quantization error distribution for float and uniform quantization grids

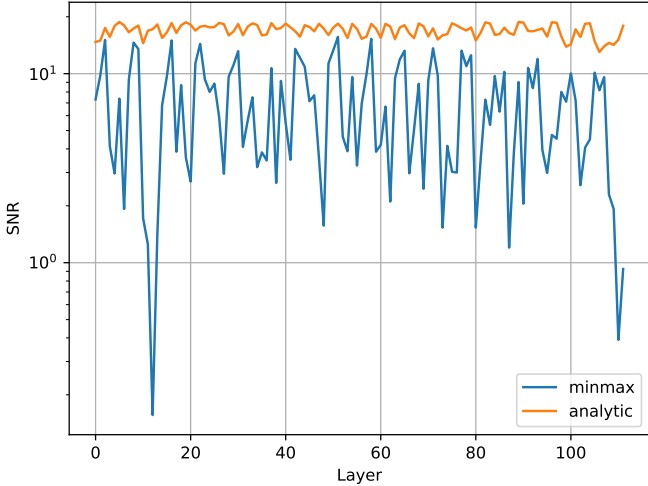

Figure 5: SNR per layer of the Llama-3-1B model under MinMax and Analytic quantization methods.

under Gaussian assumptions is a natural choice. For weights initialized from uniform distributions, the min-max method provides a better match to the initial data distribution.

Activation quantizers are initialized during the first forward pass. Since activation distributions can be highly non-Gaussian and vary across tasks and layers, the ideal approach is to fit an iterative quantizer on the first minibatch.

However, this can be computationally expensive. Empirically, we find that analytic quantizers still provide competitive initialization performance and allow stable training from the outset.

