# OpenReview forum: "StatQAT: Statistical Quantizer Optimization for Deep Networks"
_TMLR — Rejected by TMLR_

### Review · Reviewer_3aKx · 2026-04-03

**Summary Of Contributions:**

This method proposed techniques for Quantization-Aware Training (QAT) for low-bit Floating Point Quantization (FPQ) methods, such as FP4 on Blackwell hardware.

Specifically, the authors propose analytical and iterative QAT methods. These methods are applied on both Computer Vision (CV) through ResNet on CIFAR-10 as well as Natural Language Processing (NLP) tasks on several smaller LLMs such as MobileLLM and Llama.

**Additional Comments:**

References:

[1] https://arxiv.org/abs/2307.09782

[2] https://arxiv.org/abs/2503.15465

[3] https://arxiv.org/pdf/1603.05027

[4] https://www.cs.toronto.edu/~kriz/learning-features-2009-TR.pdf

[5] https://arxiv.org/abs/1905.11946

[6] https://arxiv.org/abs/2110.00476

**Audience:**

Yes

**Audience Explanation:**

The methodology of the paper and what it is trying to accomplish is of interest. The results on larger Llama LLMs are interesting as the proposed idea does well compared to baseline approaches.

**Claims And Evidence:**

No

**Claims Explanation:**

Two reasons:
1) Experimental results on CV are incredibly weak. The authors only use a very old, small model on a toy benchmark task - ResNet-18 on CIFAR-10, and they do not even cite either.
2) No hardware results showing speed-up. The purpose of quantization is to reduce hardware costs while preserving end-to-end performance. It is not enough to just show the latter.

**Requested Changes:**

1) Paper is too methodology heavy, and it does not provide a lot of intuition. Instead, the background/methodology is very dense with a lot of equations and math, but not a lot of intuition about what different components do or achieve. For instance, whenever you have an equation like (4) where you have a sigmoid summation and integrand right next to each other, you need to clearly explain why it is you're using both across the discrete and continuous domains.

2) No hardware results, see [1-2]. The paper is designed for FP4 but the experiments are conducted on V100 GPUs which are too old to run these formats, so it is difficult to see the speedup in practice.

3) The results are lacking. CV is on ResNet-18 [3] for CIFAR-10 [4] and neither are cited. Please use newer CNN models [5] or newer recipes for training ResNet [6], maybe Vision Transformer models as well for CV.

4) The presentation is lacking. Figures for ResNet are poorly formatted and have bad caption, format, so it is hard to tell what is going on just by looking at the figure. The NLP results are decent but not well positioned in the broader paper, as the table should be closer to where the text discusses its contents.

---

### Review · Reviewer_nTZV · 2026-04-07

**Summary Of Contributions:**

The work introduces a statistical framework aimed at analyzing and minimizing quantization error in both uniform and floating-point representations for neural networks. It proposes two types of quantizers (iterative and analytic) designed to adapt to data distributions during training, with particular emphasis on Gaussian assumptions for weights. These methods are integrated into quantization-aware training and evaluated across convolutional and language models. Reported results suggest improvements in stability and accuracy.

**Additional Comments:**

No further comments.

**Audience:**

Yes

**Audience Explanation:**

The paper touches relevant topics that are interesting for the TMLR community.

**Broader Impact Concerns:**

The work does not include a dedicated discussion on broader impacts.

**Claims And Evidence:**

No

**Claims Explanation:**

1.	The design decisions underlying the analytic quantizer, particularly the assumption of Gaussian-distributed weights (Section 3.1.2), are presented as broadly applicable, yet the empirical validation does not convincingly demonstrate robustness when this assumption is violated.

2.	The experimental setup described in Section 4 lacks important implementation details that would be necessary for reproducibility, including specifics on hyperparameter tuning, initialization strategies for quantizers, and computational overhead measurements.

3.	The evaluation is limited in diversity of datasets and tasks, with CIFAR-10 used for the ResNet study and WikiText-2 plus a set of reasoning benchmarks for language models, which do not fully stress the generality of the approach. Additional experiments on more varied or challenging datasets would be needed.

4.	The presentation of results in figures and tables lacks sufficient analytical discussion of the observations.  For instance, Figure 1 (page 6) and Figure 2 (page 7) are described only briefly without a structured set of observations or deeper interpretation of trends.

5.	The claimed efficiency advantages of the analytic quantizer are not rigorously quantified, since there is no explicit comparison of training time, memory overhead, or convergence behavior relative to iterative or baseline methods.

**Requested Changes:**

1.	The discussion of related work (Section 5) lacks a sufficiently critical comparison with closely related quantization-aware training approaches, particularly those that already incorporate statistical assumptions or closed-form approximations. While several methods are listed, the text does not clearly articulate what specific limitations remain unresolved and how the present framework overcomes them in a concrete and measurable way. It is recommended to discuss it clearly.

2.	The discussion of the methodology in Section 3, especially the transition from the general error formulation (Eq. 4) to the proposed iterative and analytic quantizers, is difficult to follow due to missing intermediate reasoning steps and lack of a clear top-level algorithmic. Although Algorithm 1 and Algorithm 2 are provided later (page 6), they are not sufficiently integrated with the earlier derivations, leaving ambiguity regarding which components constitute the actual contribution versus standard adaptations of k-means-like optimization. It is recommended to improve it.

3.	Based on the above comments on claims supported by evidence, several changes are required to improve these aspects.

---

### Review · Reviewer_Uw5s · 2026-04-15

**Summary Of Contributions:**

This paper introduces a statistical error-analysis framework for uniform and floating-point quantization, deriving iterative quantizers (for arbitrary distributions) and analytic quantizers (for Gaussian-like weights) with closed-form updates.
The framework is extended to FP4 formats and integrated into QAT via single-step updates.
Experiments cover ResNet-18/CIFAR-10 ablation and LLM evaluation (MobileLLM, Llama) under FP4/INT4, compared against LLM-QAT and ParetoQ.
### Strengths:
1. Timely topic with growing FP4 hardware support.
2. Unified error decomposition across uniform and float formats is clean.
3. The iterative quantizer is empirically strong. Under tensor-wise LLM settings, baselines collapse catastrophically while StatQAT-iterative stays near FP16.
4. The ResNet ablation and per-layer SNR analysis provide useful supporting evidence.

### Weaknesses:
1. The analytic quantizer fails on larger LLMs (Llama 3B/8B), yet the paper frames both methods as broadly effective.
2. Efficiency claims lack quantitative evidence.
3. Presentation has unfinished placeholders ("Appendix ??") and reference inconsistencies.
4. Baseline comparison is narrow (only LLM-QAT and ParetoQ).

**Audience:**

Yes

**Audience Explanation:**

The error-driven approach to quantizer design for hardware-constrained formats (FP4/INT4) is relevant to researchers in quantization, efficient training, and low-precision LLM deployment.

**Broader Impact Concerns:**

No specific ethical concerns

**Claims And Evidence:**

No

**Claims Explanation:**

The iterative quantizer is well-supported empirically.
However, the paper's overall claims are broader than the evidence warrants.
The conclusion states analytic quantizers achieve "performance comparable to the iterative variants at a fraction of the computational cost."
Table 1 contradicts this on larger models: Llama 3.2 3B FP4 tensor-wise analytic averages ~40.7 vs. iterative's ~53.5; Llama 3 8B INT4 tensor-wise analytic collapses to ~32.0 average / 732 perplexity.
Even channel-wise, analytic on Llama 3 8B INT4 gives ~36.7 / 81.8 perplexity.
Describing this as "scales less effectively" is a significant understatement.
The efficiency advantage is asserted but never measured, i.e., no wall-clock, memory, or FLOP comparisons are provided.
The unresolved "Appendix ??" placeholder further undermines presentation quality.
The evidence supports a narrower claim: the iterative method is strong, but the analytic method works on smaller models only.

**Requested Changes:**

### Critical:

- Narrow analytic quantizer claims. Explicitly state in the abstract, experiments, and conclusion that the analytic method is competitive only on smaller models and degrades sharply on Llama 3B/8B. "Scales less effectively" understates the problem.
Add efficiency measurements. Report wall-clock time or memory overhead for analytic vs. iterative vs. baselines. This is central to the paper's motivation.
- Fix placeholders and references. "Appendix ??" and notation inconsistencies must be resolved.
- Discuss failure cases substantively. Analyze why the analytic method fails at scale, e.g., does the Gaussian assumption break? Per-layer distributional diagnostics (kurtosis, QQ plots) on Llama 3B/8B might help.

### Non-critical:

- Include broader baselines.
- Discuss when the Gaussian weight assumption holds and when it does not in modern LLMs, beyond the brief mention of L2 regularization.

---

### Decision · Action_Editor_LwdJ · 2026-05-15

**Recommendation:** Reject

**Additional Comments:**

The decision is to reject the current submission. The main concerns are consistent across the reviews.

First, the claims about the analytic quantizer should be narrowed or better supported. Reviewer Uw5s notes that it degrades substantially on larger Llama models, and Reviewer nTZV questions whether the Gaussian assumption is sufficiently justified.

Second, the claimed efficiency advantage requires quantitative evidence. Reviewers Uw5s, nTZV, and 3aKx all request stronger evaluation of wall-clock cost, memory overhead, FLOPs, convergence behavior, or hardware-level speedup.

Third, the experimental scope should be expanded. Reviewer 3aKx specifically criticizes the limited ResNet-18/CIFAR-10 vision evaluation, and Reviewer nTZV similarly notes limited diversity across tasks and datasets.

Fourth, the presentation should be improved. Reviewer Uw5s points out placeholders and inconsistencies; Reviewer nTZV asks for clearer methodological exposition; and Reviewer 3aKx asks for more intuition behind the equations and algorithms.

A major revision should narrow the claims, add efficiency and hardware-aware evidence, expand the empirical evaluation, and improve the clarity of the methodology and analysis. No certification is recommended for the current version.

**Audience:**

Yes

**Audience Explanation:**

Yes. The paper studies FP4/INT4 quantization-aware training, which is relevant to efficient deep learning, low-precision training, and hardware-aware machine learning. All three reviewers indicate that the topic would be of interest to at least some members of the TMLR audience. The concerns are therefore about the strength of evidence, experimental breadth, and clarity, rather than about topical relevance.

**Claims And Evidence:**

No

**Claims Explanation:**

The paper addresses an important problem in low-bit quantization-aware training and proposes a statistical framework for iterative and analytic quantizer design. The reviewers agree that the topic is relevant and that the iterative StatQAT results, especially in FP4/INT4 LLM settings, are promising.

However, the current evidence does not fully support the paper’s claims. Reviewer Uw5s notes that the analytic quantizer is presented too broadly despite degrading substantially on larger Llama models. Reviewer nTZV raises a related concern that the Gaussian assumption behind the analytic quantizer is not convincingly validated. In addition, all three reviewers point out that the claimed efficiency advantage is not supported by quantitative evidence such as wall-clock time, memory overhead, FLOPs, convergence comparisons, or hardware measurements.

The experimental scope is also limited. Reviewer 3aKx emphasizes that the vision evaluation relies only on ResNet-18/CIFAR-10, while Reviewer nTZV similarly notes the lack of diversity across datasets and tasks. The presentation also needs improvement: Reviewer Uw5s identifies unresolved placeholders and inconsistencies, Reviewer nTZV asks for clearer derivations, and Reviewer 3aKx notes that the methodology is dense and lacks intuition.

Overall, while the iterative method appears promising, the current submission does not provide sufficiently convincing evidence for the full scope of its claims.

**Resubmission Of Major Revision:**

The authors may consider submitting a major revision at a later time.